Review

 

**Subject Area:**
cellular biology/molecular biology

mitosis, actin cytoskeleton, mitotic spindle, membrane trafficking, division orientation

**Authors for correspondence:**
Sara Sigismund
e-mail: sara.sigismund@ieo.it
Marina Mapelli
e-mail: marina.mapelli@ieo.it

# The crosstalk between microtubules, actin and membranes shapes cell division

Francesca Rizzelli[1], Maria Grazia Malabarba[1,2], Sara Sigismund[1,2] and Marina Mapelli[1]

[1]IEO, Istituto Europeo di Oncologia IRCCS, Milan, Italy
[2]Dipartimento di Oncologia ed Emato-oncologia, Università degli Studi di Milano, Milan, Italy

(iD) MM, 0000-0001-8502-0649

Mitotic progression is orchestrated by morphological and mechanical changes promoted by the coordinated activities of the microtubule (MT) cytoskeleton, the actin cytoskeleton and the plasma membrane (PM). MTs assemble the mitotic spindle, which assists sister chromatid separation, and contact the rigid and tensile actomyosin cortex rounded-up underneath the PM. Here, we highlight the dynamic crosstalk between MTs, actin and cell membranes during mitosis, and discuss the molecular connections between them. We also summarize recent views on how MT traction forces, the actomyosin cortex and membrane trafficking contribute to spindle positioning in isolated cells in culture and in epithelial sheets. Finally, we describe the emerging role of membrane trafficking in synchronizing actomyosin tension and cell shape changes with cell–substrate adhesion, cell–cell contacts and extracellular signalling events regulating proliferation.

## 1. Introduction

Mitotic progression is sustained by major cellular rearrangements that promote morphological features supporting faithful segregation of the genetic material and correct positioning of the daughter cells within the tissue. The actin and microtubule (MT) cytoskeleton, cell–cell adhesion and membrane dynamics are finely coordinated in space and time from mitotic entry to cytokinesis. In this review, we will present recent progress in the understanding of the mechanisms by which MTs, actin and membrane trafficking crosstalk to orchestrate mitosis, and describe how the interplay of intracellular mitotic events with cell-cell junctions and the extracellular matrix, controls tissue development and homeostasis. Our discussion will focus on findings derived from vertebrate cells in culture and in tissues, while referring occasionally to *Drosophila melanogaster* and *Caenorhabditis elegans* model systems for specific processes.

The review is organized in three parts: the first part will summarize the current knowledge on actin and MT cytoskeleton in mitosis with focus on how cortical actin and substrate adhesion contribute to spindle positioning. The second part addresses the role of endocytosis in mitosis, illustrating how the endocytic machinery assists reshaping and dynamics of the mitotic plasma membrane (PM). Finally, in the third session, we provide an overview of the interplay between mitotic cells and the surrounding tissue in terms of cell–cell contacts and extracellular matrix.

## 2. Mitosis and cytoskeleton rearrangements

The main effector of mitotic progression is the mitotic spindle, an MT-based structure that is assembled after nuclear envelope breakdown. It consists of a central spindle composed of MT bundles, known as kinetochore fibres (K-fibres), that connect poles to kinetochores (interpolar MTs connecting the

spindle poles) and astral MTs emanating from the centrosomes and protruding towards the cell periphery. The main function of the spindle is to ensure faithful segregation of the genetic material between daughter cells. However, it is becoming increasingly clear that the spindle serves other purposes, including the definition of the division plane [1]. In this section, we will summarize the current view on how the mitotic actomyosin cortex signals to the spindle apparatus throughout mitosis.

## 2.1. Actin and microtubule cytoskeleton in mitosis

Mitotic entry is characterized by a major cell shape change that reflects the reorganization of the cell cortex, defined as a thin actin network that underlies, and is tethered to, the PM [2] (figure 1a). Cortical actin filaments form a mesh cross-linked by actin-binding proteins and myosin motors conferring contractile and tensile properties to the cell surface [3,4], which responds to extracellular stress and intracellular signalling [5]. Specifically, in mitosis, the cortex becomes thinner with increased tension due to RhoA activation [6,7], thereby promoting the transition to a *rounded-up* shape (figure 1b). Rounding forces peak in prometaphase, and are maintained high till metaphase thanks to the Cdk1-mediated phosphorylation of DIAPH1 (Diaphanous Homolog 1 protein), which controls cortical actin polymerization [8]. The almost perfect spherical geometry of the cell is key for the mitotic spindle functions [9–11]. In prometaphase, the bipolar spindle is assembled and in metaphase it is positioned in the cell with the correct orientation, which, in general, is stably maintained in anaphase to pull sister chromatids apart. Both spindle orientation and chromosome separation rely on the actomyosin cortex providing a rigid scaffold that counteract the traction forces exerted on astral MTs by MT motors pulling towards the spindle poles. At cytokinesis onset, actomyosin contractility redistributes from the poles to the equatorial region of the cell generating an actomyosin flow that leads to the formation of the contractile ring [12,13] (figure 1c). What defines the localized polar release of cortical tension that establishes the cortical contractility gradient from the poles to the cell equator remains largely unclear. Evidence has been provided that also in cytokinesis there is crosstalk between the cortical actomyosin and spindle MTs that coordinates the site of furrow ingression with the spindle position [14], with mechanisms that partly involve the centralspindlin complex. Interestingly, in *Drosophila* neuroblasts, spindle-independent mechanisms also contribute to defining the cleavage furrow positioning and size asymmetry of daughter cells [15]. Whether these mechanisms are conserved in polarized systems in vertebrates is not known. Importantly, important roles for the MT-actin crosstalk have been described non only in mitosis, as recently summarized in the comprehensive review by Dogterom & Koenderink [11].

## 2.2. Adhesion in mitosis

In spite of a major mitotic reorganization of the actin cytoskeleton, recent studies in cultured cells indicate that the mitotic cortex retains a memory of the interphase organization of cell adhesion to the substrates mediated by actin-based retraction fibres. In interphase, canonical focal adhesion complexes, formed by the focal adhesion kinase (FAK), talin and paxillin,

associate with the cytoplasmic tail of the β-integrin subunit of integrin transmembrane receptors to form a signalling layer connecting the extracellular matrix to the cytoplasm [16] (figure 1a, *interphase CM adhesion complexes* box). Focal adhesion complexes were thought to disassemble in mitosis [17]. However, recent studies in HeLa cells suggest that a signalling layer of paxillin, vinculin and FAK remains under the cell body, referred to as *mitotic focal adhesion* (figure 1b, *mitotic focal adhesion complexes* box), to maintain substrate adhesion [18] (see also §4.4). Further studies showed that untransformed RPE-1 cells retain only β1-integrin adhesion, with β1-integrin localized underneath the cell body and retraction fibres, to promote spindle positioning and correct abscission [19]. These findings are consistent with *in vivo* experiments indicating that ablation of β1-integrin results in misoriented metaphases and anaphases in epithelial tissues including murine developing skin [20]. Great insights into the link between the mitotic spindle and substrate adhesion came from studies in cells cultured on adhesive micropatterns of defined shapes, pioneered by Bornens and Théry [21,22]. Elegant imaging and mechanosensing analyses conducted in these laboratories led to the discovery that the mitotic distribution of actin retraction fibres is a key predictor of the division orientation, leaving open the issue of which molecules transduce the mechanistic signals from the substrate to the spindle apparatus. Collectively, these results substantiate the notion that a memory of interphase cues remains during mitotic actomyosin reorganization and provides spatial information that guides cell division.

## 2.3. Interplay between shape, the actomyosin cortex and spindle orientation

What defines the position of the mitotic spindle, and hence of the division plane, has been object of intense investigations. Two hypotheses have been proposed as a molecular explanation of the spindle orientation. The first envisions the active contribution of force-generating complexes localized at specialized cortical regions able to exert traction forces on astral MTs to move the spindle (figure 1b). The second is a more simplistic view that assumes that the cell shape is the prominent factor determining the division orientation by compression. In fact, it is becoming clear that both cell shape and active cortical forces synergize to set the division plane, with modalities depending on the developmental stage and in response to external challenges [1]. Initial observations in artificially flattened amphibian eggs suggested that the spindle axis aligns with the longest axis of the cell, according to what is known as Hertwig's rule [23]. More sophisticated subsequent studies addressed the relevance of tension and cell shape deformation on spindle placement, revealing that in fact cell anisotropy acts as major determinant of spindle alignment [24]. Moderately anisotropic cells only partially obey the rule, with imperfect alignment of the spindle axis both in unperturbed conditions and upon mechanical cell stretching, while elongated cells favour division along the major axis. Cells in polarized epithelia undergo planar divisions, with the spindle perpendicular to the apico-basal polarity axis, and tend to follow Hertwig's rule for what concerns orientation in the anterior–posterior direction that relies on planar cell polarity proteins, such as Dishevelled and Vangl2 [25,26].

royalsocietypublishing.org/journal/rsob   Open Biol. **10**: 190314

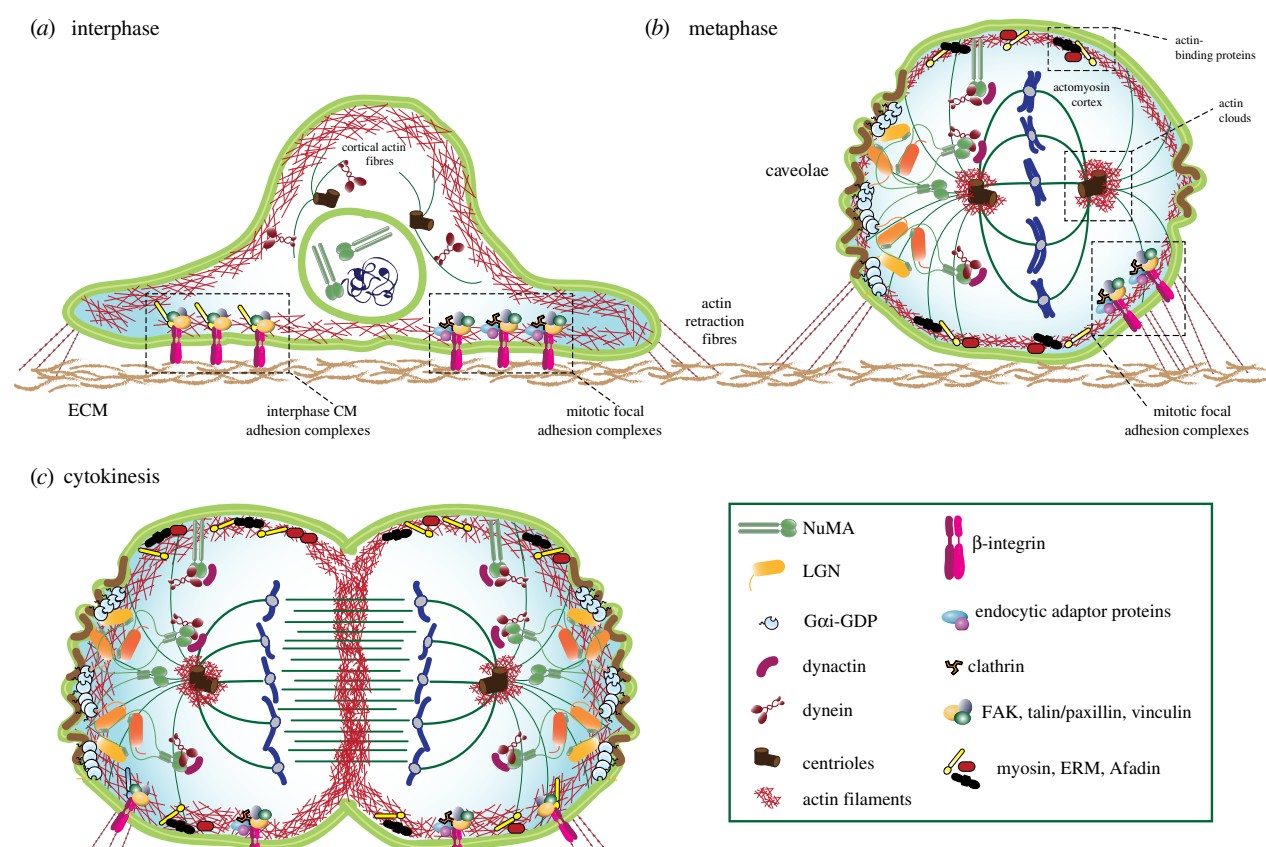

**Figure 1.** Schematic description of the organization of actin and microtubule cytoskeleton in interphase, metaphase and anaphase on vertebrate cells in culture. (*a*) In interphase cells, actin is organized in stress fibres protruding from the plasma membrane to the extracellular matrix (ECM). Cell adhesion to the substrate is mediated by focal adhesion complexes consisting of β1-integrins, the focal adhesion kinase (FAK), talin, paxillin, vinculin and clathrin (boxed in *interphase CM adhesion complexes* and *mitotic focal adhesion complexes*). Actin-associated myosin molecules confer contractility to the cortex during migration. The mitotic dynein-adaptor NuMA is nuclear in interphase. (*b*) At mitotic entry, the actin cytoskeleton is reorganized to form an isotropic contractile cortical network of actin filaments cross-linked by myosin II, which promotes a cellular morphological change known as *round-up*. Concomitantly, the canonical focal adhesion complexes present in interphase disassemble leaving *mitotic focal adhesion complexes* (boxed) containing β1-integrins and endocytic adaptors. Caveolin-1 organizes caveola-like structures at the cellular edges of retraction fibres to couple adhesion geometry to spindle positioning. After nuclear envelope break down in prometaphase, a bipolar mitotic spindle is formed by microtubules (MTs) nucleating from the two centrosomes, that capture sister chromatids at kinetochores and bring them on at the metaphase plate. In mitosis, the rigid actomyosin cortex acts as a rigid scaffold to sustain spindle positioning and elongation, thanks also to a number of cortex-associated actin-binding proteins (*actin-binding protein* box). Specifically, dynein-based MT motors are recruited localized region of the plasma membrane and exert pulling forces on astral MTs protruding from the spindle poles to the cell periphery. These force-generating machines consist of dynein/dynactin assemblies, recruited at the plasma membrane by the trimeric complex NuMA/LGN/Gαi. We recently showed that dimeric NuMA molecules assemble hetero-hexameric complexes with LGN, this way promoting the formation of cortical network of MT-motors (see also figure 2). Actin clouds distributed around the spindle pole also assist spinel positioning (*actin clouds* box). (*c*) At cytokinesis, the spindle elongates to separate sister chromatids. NuMA further enriches to the plasma membrane by direct binding to phospholipids at the polar region of the cell. Increased actomyosin cortex contractility determines the cleavage furrow ingression at the cell equator.

In response to external tension, the vertices of tight junctions (TJs) in vertebrate MDCK cells reorient and instruct the orientation axis by enriching at their site LGN, a component of the force-generating machines pulling on astral MTs (see §2.4 for a more detailed description of force-generating motors) [27]. Filming divisions in the *Drosophila* notum revealed that a similar mechanism accounts for spindle positioning also in this model system, as the LGN-binding protein, NuMA, localizes to tricellular junctions [28]. The link between TJs and spindle orientation seems to be lost during embryonic development when planar and perpendicular divisions alternate, at a given ratio, to shape tissues, as documented in the murine developing skin [20,29] and in zebrafish embryos [30].

Although these reports seem to depict TJs as the principal cues directing oriented divisions, actin has also been shown to be important, but in different ways. We already mentioned the scaffolding role of actomyosin in cell round-up. Intriguingly, in *Xenopus laevis* embryonic epithelia, actin filaments seem also to associate directly with spindle MTs [31]. In addition, the discovery of the ability of centrosomes to nucleate actin, suggested that centrosomes are the ideal hub to regulate the crosstalk between MTs and the so-called actin clouds [32] (figure 1*b*, *actin clouds* box, and figure 1*c*). Actin clouds assemble in subcortical clusters or around the centrosomes and disappear into the contractile ring in cytokinesis [33]. They have been proposed to transduce mechanical forces from the cortex to the spindle, possibly influencing spindle positioning [34–36]. If so, an interesting possibility is that asymmetric distribution of actin clouds around the mother and daughter centrosome can generate imbalanced connections of the two spindle

poles with the cortex, thereby contributing to the unequal centrosome partitioning that has been observed in cell types, such as murine neural stem cells [37] and cultured embryonic stem cells [38].

Beside actin itself, a plethora of actin-binding proteins regulates the interplay between actin and the spindle (figure 1b, *actin-binding protein* box) [39]. Cortical myosin-10 has been shown to regulate astral MT dynamics, providing a physical link between the cortex and the spindle [40], that is required for spindle orientation and acts in parallel to the LGN-dependent dynein motors [41]. In endothelial cell, myosin II has been shown to control MCAK-dependent MT growth [42]. Ezrin-radixin-moesin (ERM) proteins are membrane-actin binders that control cortical actin rigidity by cross-linking actin filaments [43]. Consistently their depletion causes membrane blebbing in *Drosophila* S2 cells and defective cell rounding *in vivo*, ultimately leading to misorientation. The orientation role of ERM proteins is conserved in vertebrate cells grown on micropatterns. Upon activation by the Ser-Thr kinase, Slik/PLKK1, ERM proteins promote cortical recruitment of LGN and NuMA [44,45]. Intriguingly, ERM proteins bind MTs, possibly contributing directly to spindle orientation [46]. We recently reported that in metaphase HeLa cells, the actin-binding protein, Afadin, controls spindle orientation by binding concomitantly to LGN and to cortical F-actin [47]. Recent data from the Williams laboratory confirmed that in murine developing skin Afadin is implicated in setting vertical and planar divisions in anaphase [48].

Beside the actin-MT cross-linkers, the number of proteins implicated in spindle positioning in vertebrate cells is steadily increasing and includes proteins involved in the regulation of astral MT-polymerization, substrate adhesion, centrosome organization, PM lipid composition and epithelial polarity. For a comprehensive review, we refer readers to the recent review by di Pietro *et al.* [1].

## 2.4. Microtubule motors moving the mitotic spindle

The functional principles of the macromolecular assemblies exerting pulling forces on astral MTs to actively move the spindle have been a subject of intense investigations. They are assembled on cytoplasmic dynein-1 (hereafter dynein) [49] and anchored at the cortex by conserved trimeric complexes consisting of the GDP-loaded Gαi subunit of heterotrimeric G-proteins, the switch protein, LGN, and the dynein-binding protein, NuMA [50] (figure 1b, and close-up in figure 2). The idea is that retrograde movement of cortically anchored dynein results in pulling forces on the spindle poles. The simplistic view of events recruiting active dynein at the cortex envisions the generation of localized Gαi-GDP pools that bind to an inhibited *closed* form of LGN inducing a conformational change compatible with NuMA binding [51]. Recently, phosphorylated LGN was shown to interact with the polarity protein DLG, further securing LGN association with the cortex in metaphase [52]. NuMA in turn recruits dynein and dynactin in a MT-independent manner [53]. Elegant optogenetic experiments by the Kyiomitsu laboratory revealed that targeting NuMA to the cortex suffices to trigger MT-pulling, while targeting dynein does not [54], suggesting that NuMA acts as a dynein-activating adaptor. This idea is corroborated by our biochemical reconstitution of the NuMA/dynein interface

showing that the N-terminal portion of NuMA contains a Hook domain and a coiled-coil region, which bind directly to the dynein light intermediate chain (Renna *et al.* 2020, unpublished data), with topologies shared by characterized dynein adaptors [55–58]. The C-terminus of NuMA harbours sites for direct binding to MTs [59–62], lipids [63,64], LGN [51,65] and 4.1R proteins [53,66] that are required for cortical actomyosin integrity, making NuMA an ideal molecule to link the mitotic PM to the spindle. Optogenetic targeting of NuMA fragments at the cortex revealed that dynein/NuMA-based force generators cluster in cortical domains visible by confocal microscopy, via an interaction module located between the NuMA coiled-coil and the LGN-binding domain [54]. In parallel, our recent structural studies showed that LGN and the C-terminus of NuMA form doughnut-shaped hetero-hexamers connected to one another by the dimeric NuMA coiled-coils, resulting in a protein network that is crucial for MT pulling [62]. The C-terminal MT-binding domain of NuMA is also required for the assembly of force generators and spindle positioning [47,54], indicating that NuMA either strengthens the anchoring of astral MTs to the PM or stabilizes dynein on astral MTs.

The view of force generators enriched cortically by Gαi-GDP/LGN/NuMA complexes leaves open the issue of what generates a localized pool of Gαi-GDP triggering the recruitment cascade. Studies in *Drosophila* neuroblasts uncovered the activity of the G-protein coupled receptor (GPCR), Tre1, in the accumulation of force generators at the apical site [67]. It is likely that still uncharacterized GPCRs exert a similar function in vertebrate systems.

Although most studies on spindle placement have focused on the LGN-mediated recruitment of NuMA, it is becoming clear that NuMA can be targeted to the PM independently of LGN. NuMA harbours a basic lipid-binding domain that is inhibited until metaphase by CDK1 phosphorylation [63,64] (figure 1c). Upon CDK1 inactivation in anaphase, NuMA is enriched at polar regions above the spindle poles by direct binding to phospholipids, which, in turn, promotes spindle elongation and sister chromatid separation. An interesting line of evidence indicates that Wnt signals can orient the division plane [38,68], possibly through the interaction of the Wnt effector Dishevelled and NuMA [25].

Together, these findings support the notion that throughout mitosis, spindle movements are orchestrated by the coordinated action of dynein-containing force generators, which are spatially organized in specific cortical regions through multivalent interactions promoted by NuMA via its ability to bind directly to MTs, lipids and 4.1R proteins.

## 3. Role of endocytosis in mitosis and cell division

In this section, we will summarize the current view on the involvement of membrane trafficking, epithelial polarity and cell–cell contacts in mitosis, and how the cellular machinery implicated in these processes communicates with the spindle apparatus. As described in the previous paragraphs, dividing cells are continuously subjected to tensile and contractile forces, which vary during the different phases of mitosis and cytokinesis, and are transduced and controlled by the actin cytoskeleton. In addition to actomyosin contractility, it is now emerging that endocytosis also has

royalsocietypublishing.org/journal/rsob    Open Biol. **10**: 190314

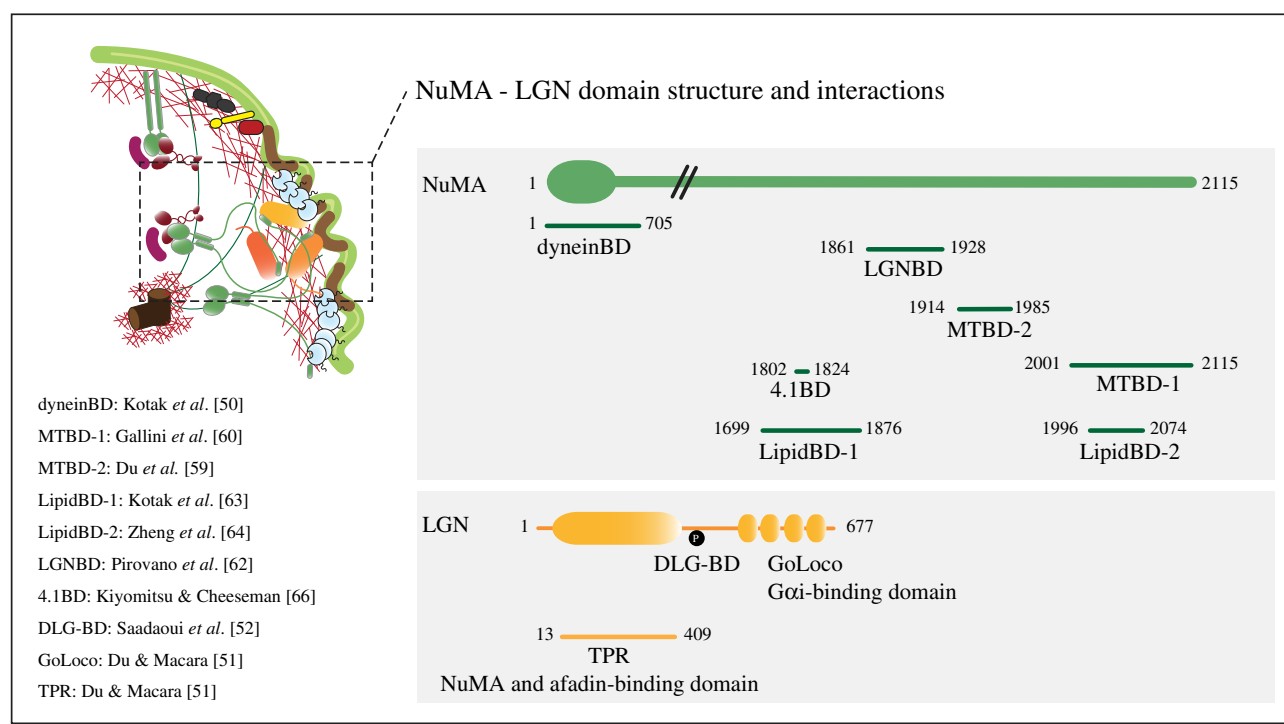

**Figure 2.** Domain structure of NuMA and LGN proteins, and mapping of their major interactions (BD, binding domain).

a critical role in PM remodelling, adherens junction (AJ) turnover and force generation in the different phases of cell division. Here, we will review evidence from the literature supporting the role of endocytosis in cell division, while we refer the reader to other more exhaustive reviews for the role of actin and the actomyosin complex [5,9,69–71]. After a brief overview of the different endocytic pathways and their relevance to PM remodelling and force generation, we will discuss the possible functions of endocytic mechanisms in mitosis, cell division and epithelial plasticity.

## 3.1. Endocytic regulation of PM remodelling and mechanical forces

Different endocytic pathways are active in different cell types, suggesting a variable impact of endocytosis on PM remodelling and mechanical forces depending on the cellular context. Endocytic pathways are broadly classified based on their dependency on the clathrin-apparatus, and thus defined as clathrin-mediated endocytosis (CME) and non-clathrin endocytosis (NCE) [72] (figure 3).

CME is active in all cell contexts although with different kinetic properties, such as lifetime and persistence of clathrin-coated pits (CCPs) [73]. In CME, the cargo is recognized by adaptor molecules—primarily AP2, but not exclusively [74–77]—that bridge the cargo to clathrin (reviewed in [78,79]). Vesicle fission is exerted by the large GTPase, dynamin (reviewed in [80]), which is also part of the scission machinery in some clathrin-independent pathways [81]. In addition, a number of reports suggest that actin plays a role in CME, facilitating PM constriction and dynamin-dependent fission. However, while this role of actin is essential in yeast cells due to the presence of the stiff yeast cell wall [82], in

mammalian cells, it appears to be relevant only when they are subjected to high membrane tension [83–85].

Two distinct types of clathrin-coated structures (CCSs) can be visualized at the PM of mammalian cells: the dynamic curved CCPs and the large, long-lived, flat clathrin lattices, called 'coated plaques' (figure 3), first observed several decades ago [85–90]. The latter structures are very stable, enriched in signalling receptors (e.g. EGFR, HGFR) and integrins [90,91]. Given these characteristics, coated plaques have been proposed to function as signalling and adhesion platforms [92]. Importantly, they assemble and expand as the rigidity of substrates increases, independently of actin and actomyosin contractility, but due to the action of αvβ5 integrin, which is particularly enriched at plaques. Importantly, αvβ5 integrin was shown to link CCSs to the substrate, in this way stabilizing them and delaying their budding from the PM, in a process termed 'frustrated endocytosis' [91]. A similar process mediated by β1-integrin has also been described for structures resembling clathrin-coated plaques present on collagen fibres (called tubular clathrin/AP2 lattices) that are critical to support 3D cell migration [93].

Coated plaques have therefore been proposed to represent a novel class of mechanosensitive stable adhesion structures, generated as a consequence of 'frustrated endocytosis' of CCSs [91]. They differ from the canonical adhesion/focal complexes that are strongly linked to the F-actin machinery and display a fast turnover and require the rapid uptake/recycling of integrins in order to allow polarization of receptors and delivery of new membrane, needed for protrusion formation and cell migration [94,95]. Instead, coated plaques seem to be strictly related—in terms of molecular composition and independence from actin—to αvβ5 integrin-enriched structures that have been involved in adhesion during mitosis [92]. Importantly, in mitosis, canonical adhesion complexes are disassembled while adhesive structures resembling plaques are maintained to preserve the interaction with the

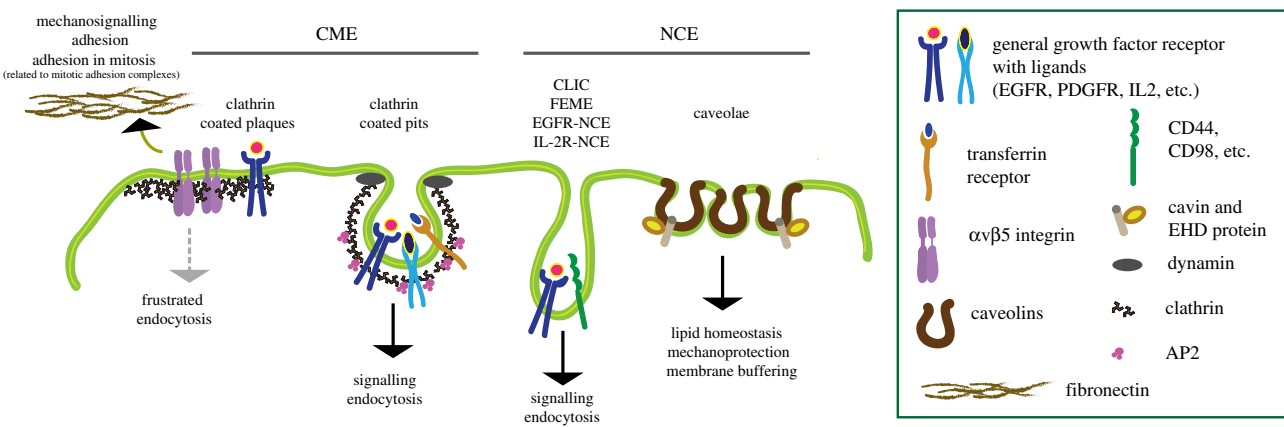

**Figure 3.** Pathways of endocytosis. Simplified schematic view of the major cellular pathways of endocytosis described in the text. Clathrin-mediated endocytosis (CME) includes clathrin-coated pits, internalizing several growth factor receptors, transferrin receptor (TfR) and others PM cargoes, and clathrin-coated plaques, enriched in αvβ5 integrin and growth factor receptors, and involved in adhesion and mechanosensitive signalling. Non-clathrin endocytosis (NCE) comprises multiple heterogeneous pathways, including clathrin-independent carriers (CLIC), fast endophilin-mediated endocytosis (FEME) and NCE pathways, involved in the internalization of the EGFR and of IL-2R, which are characterized by morphologically equivalent endocytic intermediates (i.e. tubule–vesicular invaginations) but different molecular requirements and cell context relevance. Caveolae are also a form of NCE, characterized by flask-shaped invagination enriched in caveolins and stabilizing factors/adaptors, such as cavins and EHD proteins.

substrate needed to achieve effective mitosis, daughter cell re-spreading and mitotic spindle orientation [19,96–99] (see also §4.2 and 4.3).

Differently from CME, NCE pathways include a number of heterogeneous endocytic mechanisms that are active in different cellular contexts, and which diverge at the morphological and molecular levels, their cargo and upstream regulatory signals [81]. These include, for instance, the CLIC (clathrin-independent carriers) pathway, the so-called fast endophilin-mediated endocytosis (FEME), and NCE pathways involved in the internalization of the EGFR [100] and of interleukin-2 receptor (IL-2R) [101] (figure 3). NCE pathways have been implicated in PM remodelling to different extents. For instance, the CLIC pathway is very prominent in fibroblasts where it is thought to contribute to large PM rearrangements [81,102], while FEME, given its rapid turnover at the leading edge of the cell, is predicted to have a great impact on PM remodelling during migration [83]. However, among the different NCE mechanisms, the caveolar pathway is the only one that has been directly linked to mechanosensing functions.

Caveolae are 60–80 nm diameter PM invaginations, organized in clusters or 'rosettes', which are particularly abundant on the surface of adipocytes, muscle and endothelial cells. They are very stable structures with slow turnover at the PM. Indeed, while there are few cargoes that can be internalized through caveolae, it is becoming clear that their main function is not endocytosis. Caveolae appear instead to have a critical role in lipid homeostasis and in mechanoprotection [103].

Flattening of caveolae has been observed upon osmotic swelling and cell stretching, and works as a buffering mechanism, reducing membrane tension and preventing rupture [104–107]. This function is compatible with the abundance of caveolae in tissue subjected to mechanical challenges and with their stability at steady state. Importantly, caveolae components, namely Cavins and EHD (Eps15-homology domain) proteins, have been shown to be released upon caveolae disassembly and to translocate into the nucleus where they can transduce signalling via the activation of

specific transcriptional programs [108,109]. In particular, EHD2 is critical for stabilizing caveolae structures at the PM, but it is rapidly released upon disassembly of caveolae due to mechanical stress and translocates to the nucleus where it activates the transcription of signalling effectors and cell cycle genes, as well as caveolae components themselves, to allow caveolae reconstruction after their disassembly [109].

Given the importance of endocytic pathways in the regulation of PM remodelling and lipid composition and in the buffering of mechanical forces, it is not surprising that endocytosis is tightly regulated during mitosis and cell division, and that it has been implicated in the different steps of cytokinesis, as we will discuss in the next paragraph.

## 3.2. Role of endocytosis and trafficking in the regulation of PM remodelling during mitosis

Early studies in the field of endocytosis suggested that internalization was inhibited during mitosis. Initial evidence in this direction dates back to seventies [110], when it was shown that phagocytosis and fluid-phase internalization were inhibited in mouse embryonic fibroblasts and macrophages. This was later supported by reports showing that pinocytosis [111], autophagy [112] and CCP formation were affected in mitotic cells [113,114].

Importantly, most studies pointing to endocytic arrest in mitosis were performed under conditions of mitotic synchronization, achieved using temperature shift or chemical agents, which have a strong impact on CME [115]. By contrast, experiments performed under physiological unperturbed conditions, revealed that CME proceeds during all phases of mitosis [115,116], albeit at a reduced rate. In particular, during metaphase and anaphase (figure 4a–c), a decrease in CCP density and a slowdown of CME was observed by lattice-sheet microscopy, with a recovery during cytokinesis (figure 4d) [117]. The decrease in CCP formation could be linked to actin. Indeed, the mitotic cell rounding is associated

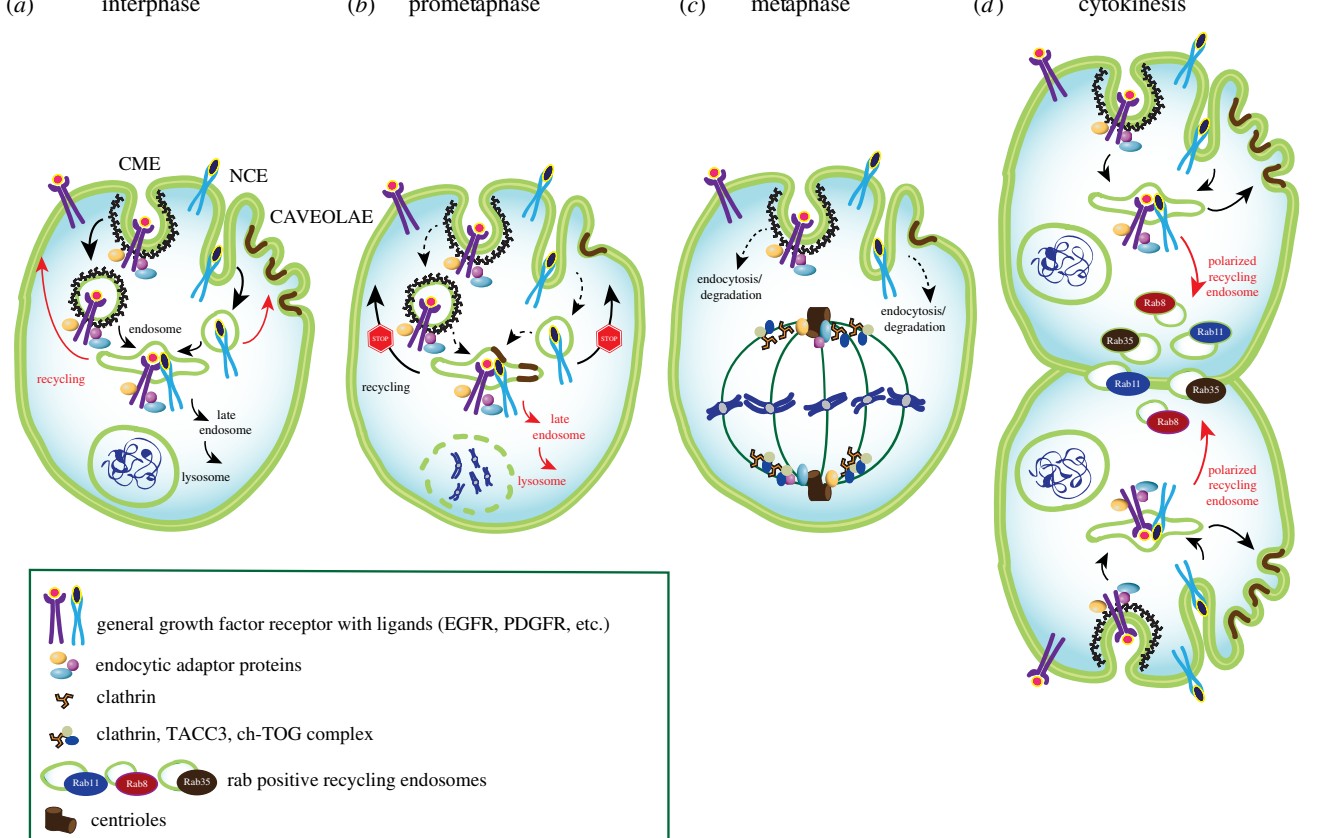

**Figure 4.** Role of endocytosis and endocytic proteins in mitosis and cell division. Regulation of the different endocytic pathways (CME, NCE and caveolae-dependent endocytosis), recycling and degradative routes in cell division. (a) During interphase, growth factor receptors, as prototype of endocytosed PM proteins, are internalized through different pathways, converged to endosome and are either recycled back to plasma membrane or destined to lysosome for degradation (according to the specific receptor, the growing conditions and the cell context). (b,c) Progressing into the different phases of mitosis, a decrease in CCP density and a slowdown of CME was observed, as well as an attenuation of NCE and a reduced number of caveolae at the PM. Recycling of internalized proteins is blocked, and degradation through lysosome is the preferred route. Caveolin-1 is redistributed to intracellular compartments. In metaphase, the tri-complex among clathrin, TACC3 and ch-TOG, which creates a novel-binding surface for MTs, is shown as an example of the 'moonlighting' function of some endocytic factors in cell division. Note that, for simplicity, clathrin is represented as a triskelion both at the PM and at the centrosomes, however in the latter case clathrin is acting as a monomer. (d) During cytokinesis, CME and NCE are fully active and recycling restarts, allowing for caveolae to come back to the cell surface. An extensive PM remodelling takes place at the furrow: Rab8, Rab11 and Rab35 regulate the polarized recycling mechanism at the cell bridge required for efficient cytokinesis.

with an increased tension of the actomyosin cortex [118], likely contrasting the invaginations of membranes occurring during endocytosis. In addition, the actin cortex thickens during mitosis to form the contractile furrow [119–121]. In parallel, recycling was also slowdown during prometaphase and metaphase (figure 4a–c) to favour cell rounding, and rescued during cytokinesis (figure 4d) to promote the increase of cell area and the subsequent flattening of cells [122].

The current view is that CME is not completely shut down during mitosis [123] and that the residual CME is critical to the internalization of specific cargoes in endosomes that are partitioned equally or asymmetrically between the two daughter cells. This is the case of the morphogen decapentaplegic (Dpp) in *Drosophila* or the planar cell polarity protein (PCP) complex in mouse that are vital to preserve tissue polarity and need to be inherited equally by daughter cells [124,125]. By contrast, the Notch receptor is internalized in SARA (smad anchor for receptor activation) endosomes that are partitioned asymmetrically and determine the different fates of the two daughter cells [126,127].

Caveolae have also been implicated in membrane remodelling during mitosis. Although there is an equilibrium between the formation and disappearance of caveolae at the PM during interphase, in mitosis, more caveolin-1 is shifted

to intracellular compartments, possibly due to the shutdown of endosomal recycling (figure 4a,b). This redistribution is reverted during cytokinesis and caveolae come back to the cell surface after anaphase (figure 4c) [128]. This behaviour suggests that caveolae dynamics might also contribute to the variation of the cell surface observed during mitosis. At the onset of mitotic cell rounding, caveolin-1 is targeted to the retracting cortical region at the proximal end of retraction fibres, where ganglioside GM1-enriched membrane domains with clusters of caveolae-like structures are formed in an integrin- and RhoA-dependent manner. Furthermore, Gαi1–LGN–NuMA, a well-known regulatory complex of spindle orientation, is targeted to the caveolin-1-enriched cortical region to guide the spindle axis towards the cellular edge retraction [129].

Finally, other NCE pathways remain active during mitosis, such as the one responsible for the uptake of the EGFR [130,131], as well as some macropinocytic events [132].

Thus, the emerging concept is that endocytosis, not only remains active during mitosis and cytokinesis, but is also crucial for the completion of these processes, because it represents, together with recycling and exocytosis, a mechanism to control membrane remodelling. Interestingly, lysosome exocytosis—a process crucially involved in PM repair [133–

royalsocietypublishing.org/journal/rsob    Open Biol. **10**: 190314

135]—has been recently shown to contribute to the increase in cell surface area when cells enter cytokinesis [136]. In particular, the last step of cytokinesis seems to require extensive PM remodelling at the furrow, which involves exocyst-mediated secretion to allow furrow contraction (figure 4d). All subunits of the exocyst complex are found at the midbody and form a ring-like structure needed for the completion of abscission. Several Rabs have been found to localize at the furrow and/or the midbody [137]. In particular, Rab11 and Rab35 regulate the recycling mechanism at the cell bridge required for efficient cytokinesis [138–141]. Similarly, Rab8-positive vesicles have been observed to be concentrated and tethered at the midbody (figure 4d) [142–144]. The fusion of these recycling endosomes is mediated by the endosomal V-SNAREs, VAMP3 and VAMP7, and their ablation inhibits the increase in surface area during telophase and leads to cell division defects [116].

Finally, lipid composition is modulated during cytokinesis, particularly at the furrow and midbody regions. Successful abscission requires phosphoinositol-3-phosphate (PI3P) production and phosphoinositol-4,5-bisphosphate (PI(4,5)P$_2$) hydrolysis [145,146]. A key function of PI3P is to recruit the protein FYVE-CENT to the bridge, which acts as a scaffold for TTC19 (tetratricopeptide repeat domain 19) [145]. As TTC19 binds to the ESCRT-III subunit, CHMP4B, it has been proposed to regulate ESCRT-III function in abscission (see §3.3). Then, prior to abscission, PI(4,5)P$_2$ is hydrolysed by the PI5 phosphatase, OCRL, which is recruited to the bridge via Rab35-positive endosomes that are recycled to the cleavage furrow [138,147].

Therefore, the balance between endocytosis and exocytosis is a fine-tune regulator of the cell surface area during division and affecting this equilibrium impairs cell rounding and cytokinesis [116,148–150].

## 3.3. Endocytic proteins with functions in mitosis and cytokinesis

Over the past decades, a number of endocytic proteins have been directly implicated in different phases of mitosis, mitotic spindle assembly and cytokinesis (table 1), independently of their role in membrane trafficking. In some cases, the molecular mechanism of action of these proteins in mitosis is equivalent to the one they exert in membrane trafficking, although in a different context. This is the case of the ESCRT-III machinery, which, through their membrane remodelling ability, have been implicated in several cellular functions, beyond multivesicular body (MVB) maturation, including cytokinesis and PM repair [152,153]. Indeed, the last phases of cytokinesis, namely the abscission phase is topologically equivalent to the membrane budding events mediated by ESCRT-III and required for intra-luminal vesicle formation at MVBs [153]. Spiral filaments of ESCRT-III have been visualized at the abscission site by electron tomography and 3D-STORM microcopy [170–172]. These filaments have been proposed to behave as elastic springs and to use the elastic energy to remodel membranes [173]. Importantly, ESCRT-III filaments at the cytokinetic abscission sites are very dynamic and are actively remodelled as cells progress through cytokinesis, in a mechanism dependent on the ATPase VPS4 [154,155]. This dynamic behaviour seems to be required to create the force necessary for membrane juxtaposition and abscission [154,155].

In other cases, endocytic proteins act in mitosis and cytokinesis completely independently of their canonical role in membrane trafficking, arguing for a true 'moonlighting'

function of these factors in cell division. This is the case, for instance, of clathrin, class II phosphoinositide 3-OH kinase α (PI3KC2α), dynamin 2, intersectin 2 and RALA-binding protein 1 (RALBP1) [156–163]. In particular, the mitotic role of clathrin has been extensively investigated. The clathrin heavy chain is recruited at the mitotic spindle of dividing cells at the entry of mitosis [154,164,165]. This pool of clathrin is not associated with membranes and its function is independent of triskelia formation. Clathrin per se has no MT-binding ability, but it forms a complex with transforming acidic coiled-coil protein 3 (TACC3) and colonic hepatic tumour overexpressed gene (ch-TOG), creating a novel-binding surface for MTs (figure 4c) [166–168]. Clathrin is critical for stabilizing MTs within the K-fibres and its depletion causes defects in chromosome separation and mitotic failure [159]. Interestingly, class II phosphoinositide 3-OH kinase α (PI3 K-C2α), an enzyme with critical role in CME, acts as a scaffold protein—independently of its kinase activity–between clathrin and TACC3 in mitosis, helping to cross-link K-fibres [169]. Downregulation of PI3 K-C2α causes spindle alterations, delayed anaphase onset and aneuploidy, indicating that a PI3 K-C2α/clathrin axis is required for genomic stability [169].

The clathrin/TACC3/ch-TOG complex was also shown to localize at the centrosome and to play a critical role in the maintenance of centrosome integrity. Interestingly, also dynamin 2 localizes at the centrosome and participates in centriole cohesion and has been implicated in the last phases of cytokinesis [162,163]. However, while the centrosomal function is due to a role of dynamin in γ-tubulin association and MT regulation, its role in cytokinesis seems to be related to its canonical membrane remodelling and fission function.

Based on these findings, it emerges that cells have adopted a strategy of using the same molecular machinery to exert different functions depending on the cell state. This is achieved by exploiting the same mechanism of action in endocytosis and in mitosis (e.g. ability to deform membranes) and/or through the acquisition of novel functions and binding abilities.

## 4. Cell division and epithelial dynamics: the role of AJs and their regulation by endocytosis

Epithelial morphogenesis represents a key process in organism shaping during development. It takes place through spatially and temporally regulated dynamic remodelling of epithelia achieved via a series of events encompassing change of cell shape and size, cell division and collective migration. In the past decade, thanks to technological advances, a growing body of evidence confirmed the impact of mechanical forces on tissue morphogenesis and epithelial plasticity [174,175].

In the process of epithelial morphogenesis, AJs—together with TJs and desmosomes—have emerged as critical regulators that sense mechanical cues, propagate signals to neighbouring cells and transduce forces into short- and long-term cellular responses [176–178]. The response of epithelia to tension by the remodelling of AJs is critical to regulate epithelial morphogenesis, tissue size and architecture in vivo. The short-term response of changes in AJ architecture is then translated into a long-term response through the activation of signalling pathways and transcriptional programs controlling proliferation, apoptosis and affecting tissue patterning [179].

**Table 1.** Summary of the endocytic proteins that are discussed in the main text and their role in mitosis and/or cytokinesis.

| endocytic protein | role in mitosis and/or cytokinesis | references |
|---|---|---|
| caveolin-1 | caveolin-1 is enriched at cortical regions, where the Gαi1–LGN–NuMA complex is targeted, to guide the spindle axis towards the cellular edge retraction; during mitosis, caveolin-1 redistributes from the plasma membrane to intracellular compartments; these changes are reversed during cytokinesis | [128,129] |
| Rab11, Rab35 | Rab11 and Rab35 regulate the recycling mechanism at the inter-cellular bridge required for efficient cytokinesis | [138–142,151] |
| Rab8 | Rab8 participates in promoting membrane addition at the cleavage furrow | [142–144,239] |
| VAMP3, VAMP7 | VAMP3 and VAMP7 mediate the fusion of the recycling endosomes to the plasma membrane; their ablation inhibits the increase in surface area during telophase and leads to cell division defects | [116] |
| OCRL | the PI5 phosphatase, OCRL, hydrolysed PI(4,5)P2 in Rab35-positive endosomes that are recycled to the cleavage furrow | [138, 147] |
| ESCRT-III machinery | ESCRT-III spiral filaments behave as elastic springs and use the elastic energy to remodel membranes<br><br>ESCRT-III complex (in particular, its subunit CHMP4B) is implicated in the abscission step of cytokinesis, together with the centrosomal scaffold protein FYVE-CENT and TTC19 | [145–147,152–155] |
| ATPase VPS4 | ATPase VPS4 participates in the remodelling of ESCRT-III filaments | [154,155] |
| intersectin 2 | intersectin 2 participates to the control of mitotic spindle orientation | [156] |
| RALBP1 | RalBP1 is involved in regulating the dynamics of the actin cytoskeleton; during mitosis RalBP1 also associates with the mitotic spindle and the centrosome, a localization that could be negatively regulated by active Ral | [157–165] |
| clathrin | the clathrin heavy chain is recruited at the mitotic spindle of dividing cells at the entry of mitosis; this function is independent of triskelia formation<br><br>clathrin, in a complex with transforming acidic coiled-coil protein 3 (TACC3) and colonic hepatic tumour overexpressed gene (ch-TOG), creates a novel-binding surface for microtubules; this complex is critical for stabilizing MTs within the K-fibres and its depletion causes defects in chromosome separation and mitotic failure | [159,166–169] |
| PI3KC2α | PI3 K-C2α acts as a scaffold protein—independently of its kinase activity—between clathrin and TACC3 in mitosis, helping to cross-link K-fibres | [169] |
| dynamin | dynamin 2 localizes at the centrosome and participates in centriole cohesion and in the last phases of cytokinesis | [162,163] |

Both the short-term and the long-term response mediated by AJs is regulated by endocytic and trafficking pathways, as we will discuss in this section.

## 4.1. AJs are critical sensors of forces in polarized epithelia

The formation of separate and specialized domains is essential to many cellular physiological processes. In epithelia, the establishment of polarity (i.e. apico-basal polarity and planar polarity) is important for the function and the integrity of tissues and consequently for organismal development [180]. Besides the polarization observed in tissue, non-polarized cells can also undergo an asymmetric distribution of biological molecules (i.e. proteins or lipids) to execute specialized functions, such as cell division, cell migration during wound healing and immune response, and degradation of the extracellular matrix. The polarity and the function of epithelia as mechanical barriers

is ensured by the cell–cell contacts [181,182]. However, cell contacts are far from being static structures: they undergo a continuous remodelling to reshape tissue architecture during development, growth and differentiation [179,180,183].

The organization of polarized epithelia in vertebrates is maintained by a tripartite junctional complex, consisting of TJs (zonula occludens), AJs (zonula adherens) and desmosomes (macula adherens) [184,185]. Desmosomes provide resilience and stability to epithelia [185], TJs regulate the passage of ions, water and macromolecules in paracellular space and establish cell polarity, and AJs are required in the very first steps of cell-cell contact formation [184].

AJs are composed of nectin-based and cadherin-based adhesions (for a review see [186,187]). The cadherin superfamily consists of diverse proteins that share a well conserved transmembrane domain and an extracellular domain containing five immunoglobulin-like repeats involved in direct interaction with cadherins on neighbouring cells. The cadherin cytoplasmic tail recruits β-catenin and p120-catenin [188]. It is through the

interaction with β-catenin that E-cadherin binds α-catenin; this interaction occurs only at cell contacts and mediates the association of AJs with the actin cytoskeleton. E-cadherin and β-catenin colocalize already in the Golgi complex and their binding is required for proper sorting of E-cadherin to AJs [189,190]. By contrast, the p120-catenin/E-cadherin association takes place at the basolateral PM, where p120-catenin stabilizes E-cadherin by preventing its endocytosis [191,192] (see also §4.2).

Multiple approaches have established in different systems that mechanical forces applied to epithelial monolayers reinforce cell–cell junctions through a positive feedback loop [178]. This reinforcement of cell contacts is based on different mechanisms involving E-cadherin and the actomyosin cytoskeleton. Application of an external force promotes the 'catch bonds' effect: this is the result of conformational changes in the interacting proteins found in AJs and/or in the actomyosin complex, which increase their affinity and the stability of the interaction [193]. For instance, E-cadherin undergoes a conformational change in its extracellular domain, thereby reinforcing homophilic interactions. This applies also to α-catenin/F-actin bonds: unfolded α-catenin stabilizes F-actin and promotes the recruitment of proteins, such as vinculin, α-actinin, formin 1 and afadin, to cell-cell junctions (reviewed in [194]). Vinculin in turn stabilizes 'open' α-catenin and triggers F-actin nucleation and actomyosin rearrangements, thus further promoting AJ reinforcement under tension [195].

The actomyosin network not only rearranges upon AJ-mediated signalling, but it is also intrinsically mechanosensitive to tension. Mechanical load is sensed by the non-muscle myosin II (MyoII), which regulates the attachment of actin to myosin heads, transforming the motor into an actin anchor thereby maintaining tension [196]. Similarly, other actin-binding proteins, such as formins and eplins, were shown to be mechanosensitive and to respond to increased tension through conformational changes, enhancing their actin polymerization ability [197–199] and inducing the polarization of actomyosin across the tissue [200].

Thus, a number of junctional components and actomyosin-binding proteins can sense mechanical cues and respond accordingly. AJs are, therefore, considered as mechanosensing and mechanotransducing platforms [193,194], able to respond to and regulate different processes involving mechanical forces, including collective cell migration, cell-to-cell intercalation and cell division [179,201]. Endocytosis is thought to be regulated in response to mechanical stimuli and to play a critical role in these different cellular processes (see, for instance, [202–204]). In the next sections, we will focus on the role of endocytosis and the endocytic machinery in AJ remodelling and in the maintenance of epithelia integrity during cell division.

## 4.2. The role of endocytosis in AJ remodelling

Endocytosis is one of the major mechanisms involved in the assembly and remodelling of AJs [192,205,206]. Immature junctions require continuous cycles of endocytosis and recycling to mature and assemble into more stable junctional structures. However, once mature, AJs are also continuously remodelled by trafficking of the component proteins (reviewed in [192,205,206]). Internalization assays in MDCK cell monolayers revealed that a small fraction of E-cadherin is constantly internalized through CME and then recycled back to AJs, and suggested the existence of a storage compartment from where E-cadherin can be rapidly recycled back to the PM

[190,207–209]. Depending on the cell context, E-cadherin has also been shown to be internalized via NCE, including dynamin-dependent mechanisms and micropinocytosis [210–212]. Despite the entry route, constitutive endocytosis seems to target E-cadherin to a recycling fate, and not to lysosomal degradation, to allow for the rapid availability of E-cadherin necessary for junction remodelling (figure 5a).

A critical regulator of E-cadherin (and VE-cadherin, the vascular endothelial specific cadherin protein) endocytosis and turnover is p120-catenin [213–216]. Indeed, its depletion causes E-cadherin/VE-cadherin internalization and degradation through a dual mechanism involving both the proteasome and the lysosome. Thus, p120 acts as a negative regulator of E-cadherin endocytosis and degradation, stabilizing AJs at the cell surface. The molecular mechanism of action of p120 is still under investigation. Structural and biochemical studies suggest that the mechanism might rely on the competitive binding between p120 and endocytic adaptors on the E-cadherin cytosolic tail [215,216].

Importantly, AJ endocytosis and turnover is finely regulated by multiple signalling pathways and it is induced when cells need to detach from the neighbouring cells, for instance, during migration and epithelial-to-mesenchymal transition (EMT), or when cells need to divide within the epithelium (figure 5b) [217–219]. Indeed, HGF and other growth factors, including FGF, EGF and VEGF, have been shown to stimulate E-cadherin (or VE-cadherin) endocytosis, disassembly of AJs and destabilization of cell-cell contacts, to allow cell scattering (in the case of HGF, see for instance [220]), migration (in the case of HGF and EGF [221,222]) or to increase endothelial permeability (in the case of VEGF [223,224]). In some cases, these stimuli cause E-cadherin relocalization and its PM depletion, without affecting its protein level, at variance with TGFβ, one of the most potent and best characterized inducers of EMT [225]. Acute stimulation of epithelial cells with TGFβ promotes E-cadherin internalization and lysosomal degradation [226,227], while prolonged stimulation induces downregulation of E-cadherin mRNA and activation of the EMT transcriptional program, including induction of EMT markers (e.g. N-cadherin and vimentin), as well as EMT transcription factors (e.g. zeb, snail and slug) [225]. These events lead to the loss of AJs and epithelial properties, and the acquisition of mesenchymal-like phenotypes.

## 4.3. The role of endocytosis in the maintenance of epithelial integrity during cell division

The maintenance of epithelial integrity requires the persistence of AJs throughout development [179,182]. Nevertheless, AJs are continuously remodelled in the epithelium and this dynamic remodelling is crucial during the division of epithelial cells within a tissue. Indeed, the disengagement of established AJs between mitotic and neighbouring cells at the cleavage furrow, and the assembly of new AJs between the two daughter cells, is crucial during epithelial cell division [179,228].

A dual mechanism controls the interaction between mitotic and neighbouring cells. On the one hand, the tensile force exerted by the actomyosin contractile ring helps to overcome the strength of interaction between mitotic and non-mitotic cells; on the other, the turnover of AJs at the furrow regulates cell-to-cell communication events during the different steps of cell division [71].

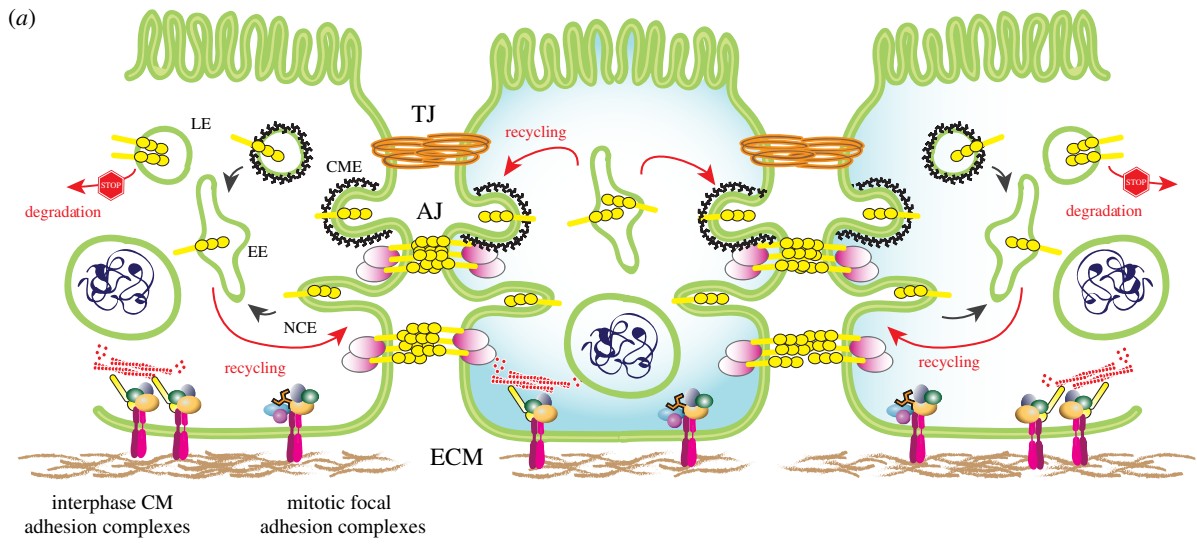

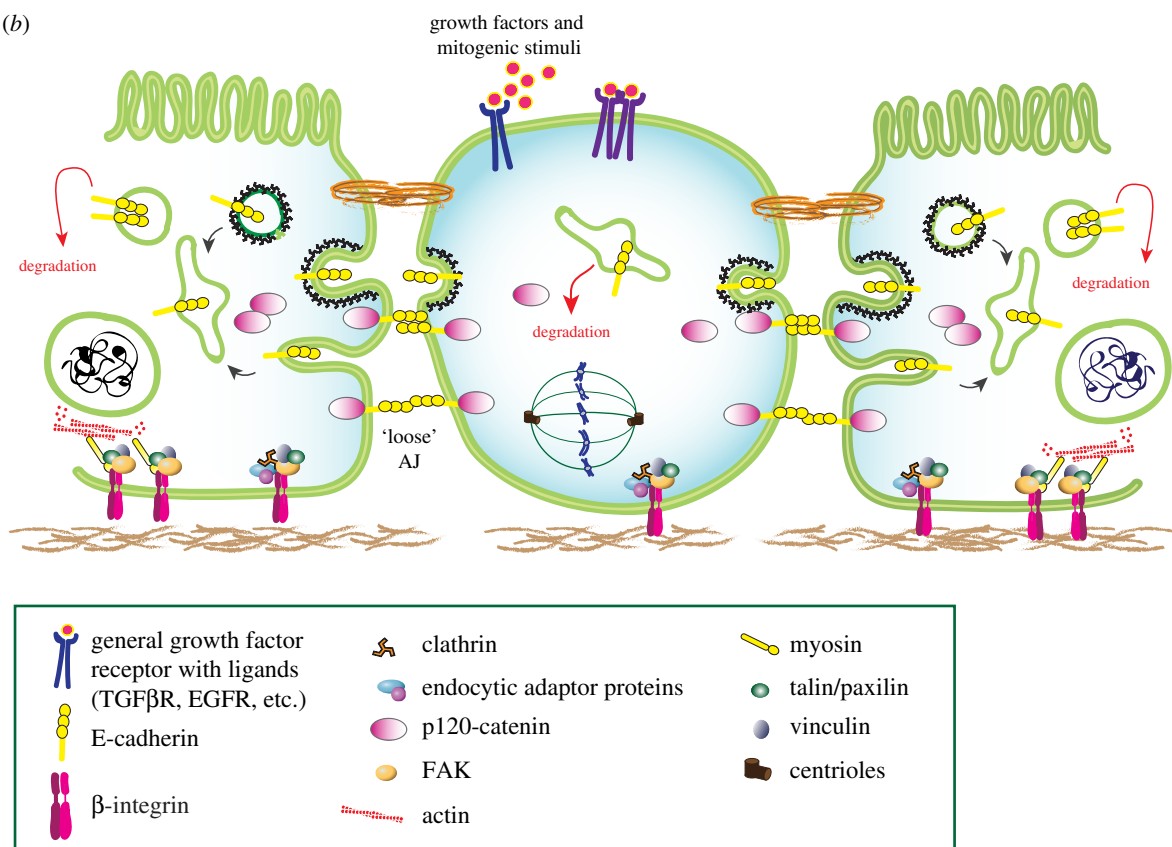

**Figure 5.** Role of endocytosis in AJ remodelling during epithelial cell division. Endocytosis regulates assembly and remodelling of AJs and, in particular, of E-cadherin. (*a*) In the epithelial monolayer, a small fraction of E-cadherin is constantly internalized and recycled back to the PM. The major described pathway of E-cadherin internalization is CME, but, depending on the cell type, it can be endocytosed also through NCE. Despite the entry route, endocytosis of E-cadherin in basal condition targets it mainly to a recycling fate (when compared with degradation), to allow the rapid availability of E-cadherin necessary for junction remodelling. In epithelial polarized cells, two type of adhesive structures are present, which connect the cell to the extracellular matrix: the cell matrix adhesions and the mitosis focal adhesions. The cell matrix (CM) adhesion complexes represent the canonical focal adhesion complexes, which links the extracellular matrix to the actin cytoskeleton through the function of myosin. The mitosis focal adhesions are devoid of myosin and therefore miss the connection to the actin cytoskeleton. These latter are the ones retained during mitosis. More recently, they have been also named reticular adhesions (RA) and shown to be related to clathrin-coated plaques. EE, early endosome; LE, late endosome. (*b*) Growth factors and mitogenic stimuli accelerate E-cadherin turnover from the PM, both in the mitotic cell as well as in the neighbouring cells, leading to E-cadherin targeting for lysosomal degradation. This causes a decrease in E-cadherin PM levels and a rearrangement of AJs that become 'loose', thus facilitating furrow ingression and cytokinesis. In the mitotic cells, only mitosis focal adhesions are retained, which provide the positional memory to the cell after cell division. These mitosis-resistant adhesion complexes are also enriched in clathrin and endocytic adaptor proteins, thus resembling the so-called clathrin-coated plaques, previously described at the basal surface of non-polarized cells.

First, AJs were shown to be critical for the asymmetric furrowing generally observed in epithelial cells. Indeed, when cells divide parallel to the plane of the epithelium, the so-called planar epithelial cell division, an unequal ingression of the cleavage furrow along the apical-basal axis is observed (e.g. in cultured MDCK cells or hepatocytes,

mouse intestine, vertebrate neuroepithelium and some *Drosophila* tissues, reviewed in [219]). This basal-to-apical asymmetric ingression of the furrow causes the apical positioning of the actomyosin contractile ring and of the midbody [10,179]. This is due to the association of the ring with AJs that are apically localized and, indeed, upon depletion of E-cadherin or β-catenin or in the presence of β-catenin mutations, the furrow becomes symmetric [229–231].

Second, AJs are important mechanotransducers that sense changes in contractility occurring during furrow ingression: they are rapidly remodelled and transduce information to the neighbouring cells [179,193]. The critical signal is the withdrawal of the membrane of the neighbouring cell, which causes a local decrease in E-cadherin levels just before the formation of the new membrane interface between the two daughter cells. The dilution of E-cadherin seems to be due to a local junction elongation determined by the pulling forces exerted by the contractile ring [232], but also to increased E-cadherin endocytosis and degradation at the interface (figure 5b). The reduction in E-cadherin levels are then sensed by the neighbouring cells and determines a self-organized actomyosin flow in the neighbouring cells that produces forces needed to re-establish cell polarity and shape. This then feedbacks on junction remodelling [202,232,233]. These observations point to the crucial role of endocytosis and trafficking in regulating AJ-dependent cell mechanics during division [234] and, indeed, it is known that E-Cadherin endocytosis remains active during mitosis both in vertebrate cells [235] and in *Drosophila* [236].

## 4.4. Interaction of mitotic cells with the extracellular matrix: unexpected link between mitotic focal adhesions and endocytic plaques

Not only is the regulation of cell–cell junctions critical to preserve tissue integrity during epithelial cell division, but also the adhesion of mitotic cells to the extracellular matrix plays a crucial role in this process [22,234,237].

As discussed in §2.2, canonical cell-matrix adhesion complexes are disassembled during mitosis, while mitosis-specific adhesion sites are maintained, providing positional memory to mitotic cells and allowing mitotic-spindle orientation, daughter cell separation and re-spreading (figure 5a,b) [96,97]. These structures have been described by different laboratories to be present in several cellular contexts and to display distinct features [19,96,98]. Despite some differences, mitosis-resistant adhesion sites are all enriched in integrins (αvβ5-integrin and/or β1-integrin), while they are devoid of classical adhesion components (such as talin or zyxin) and are completely independent of actin [19,98]. Mitotic focal adhesions present a peculiar dynamic, growing isotropically and hence are stationary, at variance with canonical interphase adhesion sites. They are thus stable structures with a slow turnover. Interestingly, the ability of cells to enter mitosis depends on substrate rigidity, as cells are unable to divide on soft substrates, and this correlates well with the growth and maturation of these mitosis focal adhesion sites [19,238] that are assembled only at an optimum stiffness (dependent on the cell type [239]).

Interestingly, a class of these adhesion complexes termed 'reticular adhesions' (RAs) because of their net-like appearance [98], are enriched in proteins involved in endocytosis and trafficking, including clathrin, AP2, eps15, Numb and others (figure 5b) [98,99]. These findings led to the intriguing

hypothesis that mitotic focal adhesion sites and clathrin-coated plaques are indeed closely related structures [92]. They are both very stable with slow turnover from the PM and composed mainly of integrins, while actin is not enriched and does not play any role in their dynamics. Additionally, both structures are regulated by the rigidity of the substrate, as they both grow and mature as the stiffness increases. Although more work is needed to clarify the relationship between clathrin-coated plaques and mitosis focal adhesions, these findings suggest an additional and novel function for the endocytic machinery in regulating forces at the PM crucial for mitosis and cell division.

## 5. Conclusion

Over the last two decades, our knowledge of the mechanisms governing mitotic progression has significantly increased. High-resolution imaging coupled with mechanotransduction assays have uncovered important connections between the functions of the mitotic spindle and the actomyosin cortex, as well as between actomyosin contractility, membrane dynamics and cell contacts with the surrounding environments. In parallel, the molecular identity of key players of mitotic processes have been discovered in endogenous settings by genome editing protocols. Collectively, these experiments have highlighted how the mitotic spindle, that has so far been regarded as the fundamental apparatus orchestrating cell division from a mechanistic standpoint, acts in a synergic manner with the actin cytoskeleton and membrane lipids throughout the different mitotic phases. A remarkable notion stemming from the most recent investigations is that the understanding of the intimate crosstalk between MTs, actin and lipids, relies on measurements of morphological cellular changes in time at a nanometer resolution. In this perspective, the recent advances in super-resolution microscopy and lattice light sheet microscopy, combined with the possibility of fluorescently tagging individual cellular components to follow their dynamics, holds great promise of being able to grasp the fine details of the events underlying mitotic progression at a molecular scale.

Finally, the study of the molecular mechanisms of mitosis has been carried out primarily in cultured cells in isolation. We believe that a major direction of future investigations will be understanding how these mechanisms adapt to sustain mitosis in tissues, under physiological conditions or in response to extracellular stimuli and challenges. In this scenario, it will be fascinating to explore how mitotic processes are regulated in stem cells to promote cell fate definition of the daughter cells during morphogenesis and regeneration. We are confident that the technological tools are now advanced enough to begin tackling these fundamental questions.

Data accessibility. This article has no additional data.

Authors' contributions. F.R. and M.G.M. helped writing the text and preparing the figures. S.S. and M.M. designed the review content and wrote the manuscript.

Competing interests. We declare we have no competing interests.

Funding. This work was supported by grant to M.M. from the Italian Association for Cancer Research (AIRC) (grant no. IG 18629) and the Ministry of Health grant no. (RF-2013-02357254). S.S. was supported by the Worldwide Cancer Research grant no. (16-1245), the PSR2018 and the PSR2019 Research Grants from the University of Milan. This work was partially supported by the Italian Ministry of Health with Ricerca Corrente and 5 × 1000 funds.

Acknowledgements. We thank Rosalind Gunby for critically reading the manuscript.

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
