## [Reviewer comments · Open Biology]

Review History

RSOB-19-0314.R0 (Original submission)

Review form: Reviewer 1

Recommendation

Accept with minor revision (please list in comments)

Do you have any ethical concerns with this paper?

No

Comments to the Author

This review discusses the current knowledge about the molecular mechanisms at the interplay between the actomyosin cortex, the microtubule spindle and membrane trafficking in the coordination of mitotic progression. This review is well written and well documented, spanning old and new bibliography, a rarity nowadays where reviews usually cover only the last 5 to 10 years. It goes deep in the understanding of the processes and is not just a list of papers.

I have only very few comments to improve it:

1/ I would reinforce the part on the interplay between actin and microtubules (end of page 6).

The authors cite the paper describing the ability of centrosomes to nucleate actin (Farina EMBO J

2019), but they should also cite these very recent other papers highlighting the fact F-actin is known to impact microtubule architecture and dynamics, potentially acting on chromosome behavior (Chaigne et al., 2016; Colin et al., 2018; Inoue et al., 2019; Farina et al., 2019; Kita et al., 2019; Plessner et al., 2019).

- Chaigne, A., Campillo, C., Voituriez, R., Gov, N.S., Sykes, C., Verlhac, M.H., Terret, M.E. 2016. F-actin mechanics control spindle centring in the mouse zygote. *Nat Commun.* 7:10253.
- Colin, A., Singaravelu, P., Théry, M., Blanchoin, L., and Gueroui, Z. 2018. Actin-Network Architecture Regulates Microtubule Dynamics. *Curr Biol.* 28(16):2647-2656.e4.
- Inoue, D., Obino, D., Pineau, J., Farina, F., Gaillard, J., Guerin, C., Blanchoin, L., Lennon-Duménil, A.M., and Théry, M. 2019. Actin filaments regulate microtubule growth at the centrosome. *EMBO J.* 38(11). pii: e99630.
- Plessner, M., Knerr, J., and Grosse, R. 2019. Centrosomal Actin Assembly Is Required for Proper Mitotic Spindle Formation and Chromosome Congression. *iScience.* 15:274-281.
- Kita, A.M., Swider, Z.T., Erofeev, I., Halloran, M.C., Goryachev, A.B., and Bement, W.M. 2019. Spindle-F-actin interactions in mitotic spindles in an intact vertebrate epithelium. *Mol Biol Cell.* 30(14):1645-1654.

2/ I would also cite this paper, showing the mechanosensitive role of cortical myosin-II on microtubule growth in endothelial cells, where inhibition of myosin-II activity prevents MCAK mediated MT growth:

- D'Angelo, L., Myer, N.M., and Myers, K.A. 2017. MCAK-mediated regulation of endothelial cell microtubule dynamics is mechanosensitive to myosin-II contractility. *Mol. Biol. Cell.* 28:1223-1237.

3/ I would cite this paper, an example of biochemical signaling between the cortex and the chromosomes, showing the coordination by Cdk1 of cortical tension maintenance and SAC inactivation at anaphase onset in mitotic cells.

- Nishimura, K., Johmura, Y., Deguchi, K., Jiang, Z., Uchida, K.S.K., Suzuki, N., Shimada, M., Chiba, Y., Hirota, T., Yoshimura, S.H., Kono, K., and Nakanishi, M. 2019. Cdk1-mediated DIAPH1 phosphorylation maintains metaphase cortical tension and inactivates the spindle assembly checkpoint at anaphase. *Nat Commun.* 10(1):981.

4/ At last, please replace Bornes and Thiery end of page 5 by Bornens and Thery.

Review form: Reviewer 2

Recommendation

Accept with minor revision (please list in comments)

Do you have any ethical concerns with this paper?

No

Comments to the Author

The review paper by Rizzelli et al. entitled 'The crosstalk between microtubules, actin and membranes shape cell division' focuses on an interesting area to review. This paper presents an overview of current studies focusing on dynamics of the actin- and microtubule cytoskeleton and their link to the plasma membrane organization upon mitosis. Finally, the authors also highlight studies focusing on the relevance of these processes for cell- substrate and cell-cell adhesion as well as ECM signaling upon mitotic cell division and cytokinesis for the case of cells being embedded in epithelial sheets.

Overall, the broader relevance of this nice review paper to the field of cell biology with a focus on

mitotic cytoskeleton and adhesion is evident and my suggestions therefore only minor but may help to improve the reach of this paper, especially regarding figures and tables, as stated below.
Abstract and Introduction

The reader would benefit from getting a complete outline of the whole manuscript here including the three sections and ten subsections – if not provided by the journal anyway. All further comments are following this outline's structure.

1) Mitosis and cytoskeleton rearrangements

1.1 Actin and microtubule cytoskeleton in mitosis

Citing thirteen paper in half a page seems a bit overwhelming, but with some more detail per paper – or less citations - it might be easier to follow the authors guide to the cited literature. For a general introduction, I would also recommend citing the paper by Dogterom and Koenderink on 'Actin-microtubule crosstalk in cell biology' for a non-mitotic view on the subject.

1.2 Adhesion in mitosis

As the figures 1 A to C burst with information, I would recommend here to have one extra 'zoom in'/focus on the comparison of mitotic focal adhesion to the interphase adhesion and change the title also to adhesion in interphase and mitosis. Otherwise, it might be more instructive to cite figure 1 A and B already here, as the link from adhesion to spindle orientation becomes clear only in the latter figure but is already mentioned in the text.

1.3 Interplay between shape, the actomyosin cortex and spindle orientation

The sentence containing first LGN mentioning is a bit misleading because of coma or missing parenthesis right before the word LGN. Also the general threefold subdivision in LGN and NuMA, actin clouds and ABP like myosin, dynein, ERM or Afdain might be prepared more by numbering or subsectioning components in figure 1 B. Again starting to mention the next topic from section 1.4, the MT motors, already in this subsection is a bit confusing (as with the spindle orientation in section 1.2 and 1.3).

1.4 Microtubule motors moving the mitotic spindle

It might be enough to refer to Figure 1 C only in this subsection and the also show Gai only here and not already in figure 1B. On the other hand, the reader might benefit from a molecular explanation of the structure of NuMA or/and 4.1R proteins in advance, possibly by having another 'zoom in' in figure 1C on its interaction with Gai, dynactin and dynein. As the authors cite roughly 20 paper in sections 1.3. and 1.4, these resemble highlight parts and might therefore be reflected a bit more also by the figures.

2) Role of endocytosis in mitosis and cell division

2.1 Endocytic regulation of PM remodeling and mechanical forces

In this chapter, there is no reference to a figure although I would suggest that a figure detailing the difference between Clathrin pits and plaques – especially in terms of integrins and mitosis relevance - would be helpful to understand the term 'frustrated endocytosis'. The same is valid for the distinction between CME and NCE including caveolae, thus resembling a figure 2A with a focus on the internalization routes at the top. The authors cite 35 paper in this subchapter, so an additional (sub)figure would be good to understand the context.

2.2 Role of endocytosis and trafficking in the regulation of PM remodeling during mitosis

Besides internalization via CME, caveolae and NCE, recycling, (lysosome) exocytosis as well as membrane composition modulation are introduced here as cell surface are regulators. I would suggest to try to make a table for this very comprehensive subchapter (60 papers with 2.3) – the probably most central one in this review - summarizing some of the central results found for the actors like the different Rab proteins, but possibly also some of the actors mentioned in section 2.3.

2.3 Endocytic proteins with functions in mitosis and cytokinesis

See above, point 2.2

3) Cell division and epithelial dynamics: the role of AJs and their regulation by endocytosis

3.1 AJs are critical sensors of forces in polarized epithelia

A more detailed exemplary scheme of the e-cadherine might be show in figure 3 A but is not absolutely relevant for the reader to understand the dynamics summarized in section 3.

However, I would include the part on EMT as one example here and only start the next subsection with the sentence including literature citations 168 and 171, as the mitotic focus becomes more obvious here.

3.2 The role of endocytosis in AJ remodeling during epithelial cell division

See above. There are some formatting issues before citation 218. EE and LE is not defined in figure 3 A nor in the caption.

3.3 Interaction of mitotic cells with the extracellular matrix:

unexpected link between mitotic focal adhesion and endocytic plaques

I suggest to label the reticular adhesion in figure 3 B with the abbreviation RA introduced in the text.

Decision letter (RSOB-19-0314.R0)

27-Jan-2020

Dear Dr Mapelli

We are pleased to inform you that your manuscript RSOB-19-0314 entitled "The crosstalk between microtubules, actin and membranes shape cell division" has been accepted by the Editor for publication in Open Biology. The reviewer(s) have recommended publication, but also suggest some minor revisions to your manuscript. Therefore, we invite you to respond to the reviewer(s)' comments and revise your manuscript.

Please submit the revised version of your manuscript within 14 days. If you do not think you will be able to meet this date please let us know immediately and we can extend this deadline for you.

1) A text file of the manuscript (doc, txt, rtf or tex), including the references, tables (including captions) and figure captions. Please remove any tracked changes from the text before submission. PDF files are not an accepted format for the "Main Document".

2) A separate electronic file of each figure (tiff, EPS or print-quality PDF preferred). The format should be produced directly from original creation package, or original software format. Please note that PowerPoint files are not accepted.

3) Electronic supplementary material: this should be contained in a separate file from the main text and meet our ESM criteria (see <http://royalsocietypublishing.org/instructions-authors#question5>). All supplementary materials accompanying an accepted article will be treated as in their final form. They will be published alongside the paper on the journal website and posted on the online figshare repository. Files on figshare will be made available approximately one week before the accompanying article so that the supplementary material can be attributed a unique DOI.

Online supplementary material will also carry the title and description provided during submission, so please ensure these are accurate and informative. Note that the Royal Society will not edit or typeset supplementary material and it will be hosted as provided. Please ensure that the supplementary material includes the paper details (authors, title, journal name, article DOI). Your article DOI will be 10.1098/rsob.2016[*last 4 digits of e.g. 10.1098/rsob.20160049*].

4) A media summary: a short non-technical summary (up to 100 words) of the key findings/importance of your manuscript. Please try to write in simple English, avoid jargon, explain the importance of the topic, outline the main implications and describe why this topic is newsworthy.

Images

Data-Sharing

It is a condition of publication that data supporting your paper are made available. Data should be made available either in the electronic supplementary material or through an appropriate repository. Details of how to access data should be included in your paper. Please see <http://royalsocietypublishing.org/site/authors/policy.xhtml#question6> for more details.

Data accessibility section

Sincerely,

The Open Biology Team

<mailto:openbiology@royalsociety.org>

Reviewer(s)' Comments to Author:

Referee: 1

Comments to the Author(s)

This review discusses the current knowledge about the molecular mechanisms at the interplay between the actomyosin cortex, the microtubule spindle and membrane trafficking in the

coordination of mitotic progression. This review is well written and well documented, spanning old and new bibliography, a rarity nowadays where reviews usually cover only the last 5 to 10 years. It goes deep in the understanding of the processes and is not just a list of papers.

I have only very few comments to improve it:

1/ I would reinforce the part on the interplay between actin and microtubules (end of page 6). The authors cite the paper describing the ability of centrosomes to nucleate actin (Farina EMBO J 2019), but they should also cite these very recent other papers highlighting the fact F-actin is known to impact microtubule architecture and dynamics, potentially acting on chromosome behavior (Chaigne et al., 2016; Colin et al., 2018; Inoue et al., 2019; Farina et al., 2019; Kita et al., 2019; Plessner et al., 2019).

- Chaigne, A., Campillo, C., Voituriez, R., Gov, N.S., Sykes, C., Verlhac, M.H., Terret, M.E. 2016. F-actin mechanics control spindle centring in the mouse zygote. *Nat Commun.* 7:10253.

- Colin, A., Singaravelu, P., Théry, M., Blanchoin, L., and Gueroui, Z. 2018. Actin-Network Architecture Regulates Microtubule Dynamics. *Curr Biol.* 28(16):2647-2656.e4.

- Inoue, D., Obino, D., Pineau, J., Farina, F., Gaillard, J., Guerin, C., Blanchoin, L., Lennon-Duménil, A.M., and Théry, M. 2019. Actin filaments regulate microtubule growth at the centrosome. *EMBO J.* 38(11). pii: e99630.

- Plessner, M., Knerr, J., and Grosse, R. 2019. Centrosomal Actin Assembly Is Required for Proper Mitotic Spindle Formation and Chromosome Congression. *iScience.* 15:274-281.

- Kita, A.M., Swider, Z.T., Erofeev, I., Halloran, M.C., Goryachev, A.B., and Bement, W.M. 2019. Spindle-F-actin interactions in mitotic spindles in an intact vertebrate epithelium. *Mol Biol Cell.* 30(14):1645-1654.

2/ I would also cite this paper, showing the mechanosensitive role of cortical myosin-II on microtubule growth in endothelial cells, where inhibition of myosin-II activity prevents MCAK mediated MT growth:

- D'Angelo, L., Myer, N.M., and Myers, K.A. 2017. MCAK-mediated regulation of endothelial cell microtubule dynamics is mechanosensitive to myosin-II contractility. *Mol. Biol. Cell.* 28:1223-1237.

3/ I would cite this paper, an example of biochemical signaling between the cortex and the chromosomes, showing the coordination by Cdk1 of cortical tension maintenance and SAC inactivation at anaphase onset in mitotic cells.

- Nishimura, K., Johmura, Y., Deguchi, K., Jiang, Z., Uchida, K.S.K., Suzuki, N., Shimada, M., Chiba, Y., Hirota, T., Yoshimura, S.H., Kono, K., and Nakanishi, M. 2019. Cdk1-mediated DIAPH1 phosphorylation maintains metaphase cortical tension and inactivates the spindle assembly checkpoint at anaphase. *Nat Commun.* 10(1):981.

4/ At last, please replace Bornes and Thiery end of page 5 by Bornens and Thiery.

Referee: 2

Comments to the Author(s)

The review paper by Rizzelli et al. entitled 'The crosstalk between microtubules, actin and membranes shape cell division' focuses on an interesting area to review. This paper presents an overview of current studies focusing on dynamics of the actin- and microtubule cytoskeleton and their link to the plasma membrane organization upon mitosis. Finally, the authors also highlight studies focusing on the relevance of these processes for cell- substrate and cell-cell adhesion as well as ECM signaling upon mitotic cell division and cytokinesis for the case of cells being embedded in epithelial sheets.

Overall, the broader relevance of this nice review paper to the field of cell biology with a focus on mitotic cytoskeleton and adhesion is evident and my suggestions therefore only minor but may help to improve the reach of this paper, especially regarding figures and tables, as stated below.

Abstract and Introduction

The reader would benefit from getting a complete outline of the whole manuscript here including the three sections and ten subsections – if not provided by the journal anyway. All further comments are following this outline's structure.

1) Mitosis and cytoskeleton rearrangements

1.1 Actin and microtubule cytoskeleton in mitosis

Citing thirteen paper in half a page seems a bit overwhelming, but with some more detail per paper – or less citations - it might be easier to follow the authors guide to the cited literature. For a general introduction, I would also recommend citing the paper by Dogterom and Koenderink on 'Actin-microtubule crosstalk in cell biology' for a non-mitotic view on the subject.

1.2 Adhesion in mitosis

As the figures 1 A to C burst with information, I would recommend here to have one extra 'zoom in' / focus on the comparison of mitotic focal adhesion to the interphase adhesion and change the title also to adhesion in interphase and mitosis. Otherwise, it might be more instructive to cite figure 1 A and B already here, as the link from adhesion to spindle orientation becomes clear only in the latter figure but is already mentioned in the text.

1.3 Interplay between shape, the actomyosin cortex and spindle orientation

The sentence containing first LGN mentioning is a bit misleading because of coma or missing parenthesis right before the word LGN. Also the general threefold subdivision in LGN and NuMA, actin clouds and ABP like myosin, dynein, ERM or Afdain might be prepared more by numbering or subsectioning components in figure 1 B. Again starting to mention the next topic from section 1.4, the MT motors, already in this subsection is a bit confusing (as with the spindle orientation in section 1.2 and 1.3).

1.4 Microtubule motors moving the mitotic spindle

It might be enough to refer to Figure 1 C only in this subsection and the also show Gai only here and not already in figure 1B. On the other hand, the reader might benefit from a molecular explanation of the structure of NuMA or/and 4.1R proteins in advance, possibly by having another 'zoom in' in figure 1C on its interaction with Gai, dynactin and dynein. As the authors cite roughly 20 paper in sections 1.3. and 1.4, these resemble highlight parts and might therefore be reflected a bit more also by the figures.

2) Role of endocytosis in mitosis and cell division

2.1 Endocytic regulation of PM remodeling and mechanical forces

In this chapter, there is no reference to a figure although I would suggest that a figure detailing the difference between Clathrin pits and plaques – especially in terms of integrins and mitosis relevance - would be helpful to understand the term 'frustrated endocytosis'. The same is valid for the distinction between CME and NCE including caveolae, thus resembling a figure 2A with a focus on the internalization routes at the top. The authors cite 35 paper in this subchapter, so an additional (sub)figure would be good to understand the context.

2.2 Role of endocytosis and trafficking in the regulation of PM remodeling during mitosis

Besides internalization via CME, caveolae and NCE, recycling, (lysosome) exocytosis as well as membrane composition modulation are introduced here as cell surface are regulators. I would suggest to try to make a table for this very comprehensive subchapter (60 papers with 2.3) – the probably most central one in this review - summarizing some of the central results found for the actors like the different Rab proteins, but possibly also some of the actors mentioned in section 2.3.

2.3 Endocytic proteins with functions in mitosis and cytokinesis

See above, point 2.2

3) Cell division and epithelial dynamics: the role of AJs and their regulation by endocytosis

3.1 AJs are critical sensors of forces in polarized epithelia

A more detailed exemplary scheme of the e-cadherine might be show in figure 3 A but is not absolutely relevant for the reader to understand the dynamics summarized in section 3.

However, I would include the part on EMT as one example here and only start the next subsection with the sentence including literature citations 168 and 171, as the mitotic focus becomes more obvious here.

3.2 The role of endocytosis in AJ remodeling during epithelial cell division

See above. There are some formatting issues before citation 218. EE and LE is not defined in figure 3 A nor in the caption.

3.3 Interaction of mitotic cells with the extracellular matrix:

unexpected link between mitotic focal adhesion and endocytic plaques

I suggest to label the reticular adhesion in figure 3 B with the abbreviation RA introduced in the text.

Author's Response to Decision Letter for (RSOB-19-0314.R0)

See Appendix A.

Decision letter (RSOB-19-0314.R1)

18-Feb-2020

Dear Dr Mapelli

We are pleased to inform you that your manuscript entitled "The crosstalk between microtubules, actin and membranes shapes cell division" has been accepted by the Editor for publication in Open Biology.

Sincerely,
The Open Biology Team
mailto: openbiology@royalsociety.org

Appendix A

Department of Experimental Oncology
European Institute of Oncology
Via Adamello 16, Milan
I-20139, Italy

Marina Mapelli, PhD
+39.02.94375018
marina.mapelli@ieo.it

David M. Glover FRS
Editor in Chief
Royal Society's Open Biology

February 17th, 2020

Dear Dr. Glover,

please find enclosed the response to the Reviewers' comments to the manuscript 'The crosstalk between microtubules, actin and membranes shape cell division', which was assessed by Open Biology Referees (RSOB-19-0314).

We have revised the text and the figures of the manuscript according to the Referees' suggestions, as detailed below in the point-by-point answers to the Referees' comments.

I thank you in advance for the editorial help and the patience, and I look forward to hearing from you.

With my kindest regards,
Marina Mapelli

Reviewer(s)' Comments to Author:

Referee:

1

Comments to the Author(s)

This review discusses the current knowledge about the molecular mechanisms at the interplay between the actomyosin cortex, the microtubule spindle and membrane trafficking in the coordination of mitotic progression. This review is well written and well documented, spanning old and new bibliography, a rarity nowadays where reviews usually cover only the last 5 to 10 years. It goes deep in the understanding of the processes and is not just a list of papers.

I have only very few comments to improve it:

1/ I would reinforce the part on the interplay between actin and microtubules (end of page 6). The authors cite the paper describing the ability of centrosomes to nucleate actin (Farina

EMBO J 2019), but they should also cite these very recent other papers highlighting the fact F-actin is known to impact microtubule architecture and dynamics, potentially acting on chromosome behavior (Chaigne et al., 2016; Colin et al., 2018; Inoue et al., 2019; Farina et al., 2019; Kita et al., 2019; Plessner et al., 2019).

- Chaigne, A., Campillo, C., Voituriez, R., Gov, N.S., Sykes, C., Verlhac, M.H., Terret, M.E. 2016. F-actin mechanics control spindle centring in the mouse zygote. *Nat Commun.* 7:10253.
- Colin, A., Singaravelu, P., Théry, M., Blanchoin, L., and Gueroui, Z. 2018. Actin-Network Architecture Regulates Microtubule Dynamics. *Curr Biol.* 28(16):2647-2656.e4.
- Inoue, D., Obino, D., Pineau, J., Farina, F., Gaillard, J., Guerin, C., Blanchoin, L., Lennon-Duménil, A.M., and Théry, M. 2019. Actin filaments regulate microtubule growth at the centrosome. *EMBO J.* 38(11). pii: e99630.
- Plessner, M., Knerr, J., and Grosse, R. 2019. Centrosomal Actin Assembly Is Required for Proper Mitotic Spindle Formation and Chromosome Congression. *iScience.* 15:274-281.
- Kita, A.M., Swider, Z.T., Erofeev, I., Halloran, M.C., Goryachev, A.B., and Bement, W.M. 2019. Spindle-F-actin interactions in mitotic spindles in an intact vertebrate epithelium. *Mol Biol Cell.* 30(14):1645-1654.

We thank the Reviewer for the positive comments and the suggestions. We have added the mentioned references and strengthened the description of the mitotic interplay between actin and microtubules in the first section.

2/ I would also cite this paper, showing the mechanosensitive role of cortical myosin-II on microtubule growth in endothelial cells, where inhibition of myosin-II activity prevents MCAK mediated MT growth:

- D'Angelo, L., Myer, N.M., and Myers, K.A. 2017. MCAK-mediated regulation of endothelial cell microtubule dynamics is mechanosensitive to myosin-II contractility. *Mol. Biol. Cell.* 28:1223–1237.

We have added a sentence describing the role of myosin II in modulating MCAK-dependent microtubule growth in mitosis, and cited the corresponding reference (see page 6).

3/ I would cite this paper, an example of biochemical signaling between the cortex and the chromosomes, showing the coordination by Cdk1 of cortical tension maintenance and SAC inactivation at anaphase onset in mitotic cells.

- Nishimura, K., Johmura, Y., Deguchi, K., Jiang, Z., Uchida, K.S.K., Suzuki, N., Shimada, M., Chiba, Y., Hirota, T., Yoshimura, S.H., Kono, K., and Nakanishi, M. 2019. Cdk1-mediated DIAPH1 phosphorylation maintains metaphase cortical tension and inactivates the spindle assembly checkpoint at anaphase. *Nat Commun.* 10(1):981.

We thank the Referee for the suggestion. We have added a sentence describing the role of the Cdk1-dependent phosphorylation of DIAPH1 in ensuing and maintaining cortical tension during mitotic round up and cited the associated reference (page 4).

4/ At last, please replace Bornes and Thiery end of page 5 by Bornens and Thiery.

We have corrected the typo.

Referee-2:

Comments to the Author(s)

The review paper by Rizzelli et al. entitled ‘The crosstalk between microtubules, actin and membranes shape cell division’ focuses on an interesting area to review. This paper presents an overview of current studies focusing on dynamics of the actin- and microtubule cytoskeleton and their link to the plasma membrane organization upon mitosis. Finally, the authors also highlight studies focusing on the relevance of these processes for cell- substrate and cell-cell adhesion as well as ECM signaling upon mitotic cell division and cytokinesis for the case of cells being embedded in epithelial sheets.

Overall, the broader relevance of this nice review paper to the field of cell biology with a focus on mitotic cytoskeleton and adhesion is evident and my suggestions therefore only minor but may help to improve the reach of this paper, especially regarding figures and tables, as stated below.

Abstract and Introduction

The reader would benefit from getting a complete outline of the whole manuscript here including the three sections and ten subsections – if not provided by the journal anyway. All further comments are following this outline’s structure.

We thank the Reviewer for the suggestion. We have added a paragraph at the end of the Introduction to outline the organization of the Review in three parts, indicating the focus of the each of them. However, we have preferred to omit the list of sub-section to avoid being too detailed.

1) Mitosis and cytoskeleton rearrangements

1.1 Actin and microtubule cytoskeleton in mitosis

Citing thirteen paper in half a page seems a bit overwhelming, but with some more detail per paper – or less citations - it might be easier to follow the authors guide to the cited literature. For a general introduction, I would also recommend citing the paper by Dogterom and Koenderink on ‘Actin–microtubule crosstalk in cell biology’ for a non-mitotic view on the subject.

We agree with the Referee that the paragraph describing the actin-microtubule crosstalk is rather dense, and contains a substantial number of citations. We have better described the actin-dependent rounding mechanisms (including the role of DIAPH1), and cited the review by Dogterom and Koenderink for a comprehensive description of the current knowledge on the actin-microtubule cross-talk.

1.2 Adhesion in mitosis

As the figures 1 A to C burst with information, I would recommend here to have one extra ‘zoom in’/focus on the comparison of mitotic focal adhesion to the interphase adhesion and change the title also to adhesion in interphase and mitosis. Otherwise, it might be more instructive to cite figure 1 A and B already here, as the link from adhesion to spindle orientation becomes clear only in the latter figure but is already mentioned in the text.

We take the Referee’s point that in the paragraph “Adhesion in mitosis” we provide a snapshot of adhesion mechanisms, including interphase and mitotic adhesion complexes, that is very concise. Our decision to adopt this perspective stems from the idea that the focus of the Review is to describe changes between interphase and mitosis, rather than to provide an exhaustive description of the two cell cycle phases. Also, we are mentioning adhesion complexes mainly to connect them to the spindle positioning mechanisms described in Figure 1B. Following the Referee suggestion, to make the text clearer to the readers, we have cited Figure 1A and 1B directly in this “Adhesion in mitosis” paragraph.

1.3 Interplay between shape, the actomyosin cortex and spindle orientation

The sentence containing first LGN mentioning is a bit misleading because of coma or missing parenthesis right before the word LGN.

We have corrected the typo and removed the extra comma.

Also the general threefold subdivision in LGN and NuMA, actin clouds and ABP like myosin, dynein, ERM or Afadin might be prepared more by numbering or subsectioning components in figure 1 B.

We agree with the Reviewer. We have added dashed boxes to the Figure 1B, highlighting the modules cited by the Referee, i.e. ERM-containing mitotic focal adhesion complexes, actin clouds, and actin binding proteins, and cited them appropriately in the main text. In addition, we have added a separate close-up view of the LGN/NuMA module in an additional BOX-1.

Again starting to mention the next topic from section 1.4, the MT motors, already in this subsection is a bit confusing (as with the spindle orientation in section 1.2 and 1.3).

We thank the Referee for the comment. Our decision to start mentioning MT-motors in this paragraph reflects the notion that actin-binding proteins are implicated in mitotic cortical organization as well as in spindle placement. Thus, we preferred to introduce them already in this paragraph, and describe in detail their functions in the dedicated paragraph “*Microtubule motors moving the mitotic spindle*”.

1.4 Microtubule motors moving the mitotic spindle

It might be enough to refer to Figure 1 C only in this subsection and the also show Gai only here and not already in figure 1B. On the other hand, the reader might benefit from a molecular explanation of the structure of NuMA or/and 4.1R proteins in advance, possibly by having another ‘zoom in’ in figure 1C on its interaction with Gai, dynactin and dynein. As the authors cite roughly 20 paper in sections 1.3. and 1.4, these resemble highlight parts and might therefore be reflected a bit more also by the figures.

We thank the Referee for the suggestions. Gai molecules are implicated in recruiting LGN and NuMA at the cortex from metaphase on, therefore we prefer to leave them also in Figure 1B rather than introducing them only in Figure 1C.

We agree that the description of the properties of NuMA and LGN only in the text and references may be difficult to visualize in the context of the cartoons shown in Figure 1B-C. To facilitate the readers, we have added a BOX-1 with a close-up view of Gai/LGN/NuMA interactions, and a schematic representation of the domain structure of NuMA and LGN annotated with all the interactions mentioned in the text and the relative reference.

2) Role of endocytosis in mitosis and cell division

2.1 Endocytic regulation of PM remodeling and mechanic forces.

In this chapter, there is no reference to a figure although I would suggest that a figure detailing the difference between Clathrin pits and plaques – especially in terms of integrins and mitosis relevance - would be helpful to understand the term ‘frustrated endocytosis’. The same is valid for the distinction between CME and NCE including caveolae, thus resembling a figure 2A with a focus on the internalization routes at the top. The authors cite 35 paper is this subchapter, so an additional (sub)figure would be good to understand the context.

We agree with the Reviewer. We have added a new Figure, Figure 2, describing the different entry routes discussed in the text, with a focus to the distinction between clathrin-coated pits and plaques.

2.2 Role of endocytosis and trafficking in the regulation of PM remodeling during mitosis

Besides internalization via CME, caveolae and NCE, recycling, (lysosome) exocytosis as well as membrane composition modulation are introduced here as cell surface are regulators. I would suggest to try to make a table for this very comprehensive subchapter (60 papers with 2.3) – the probably most central one in this review - summarizing some of the central results found for the actors like the different Rab proteins, but possibly also some of the actors mentioned in section 2.3.

We thank the Reviewer for the suggestion. We have added a new Table I, summarizing the different endocytic proteins that have been discussed along the text.

2.3 Endocytic proteins with functions in mitosis and cytokinesis

See above, point 2.2

3) Cell division and epithelial dynamics: the role of AJs and their regulation by endocytosis

3.1 AJs are critical sensors of forces in polarized epithelia

A more detailed exemplary scheme of the e-cadherine might be show in figure 3 A but is not absolutely relevant for the reader to understand the dynamics summarized in section 3. However, I would include the part on EMT as one example here and only start the next subsection with the sentence including literature citations 168 and 171, as the mitotic focus becomes more obvious here.

We thank the Reviewer for the suggestion. We have rearranged the text by adding a new subsection, which starts with the sentence including literature citation 168 and 171 (now 178-181), to better highlight the mitotic focus of this paragraph.

3.2 The role of endocytosis in AJ remodeling during epithelial cell division

See above. There are some formatting issues before citation 218.

We have solved the formatting issue.

EE and LE is not defined in figure 3 A nor in the caption.

We apologize for not having clarified this in the previous version of the manuscript. We have now explained the acronyms in the figure legend (now Figure 4A).

3.3 Interaction of mitotic cells with the extracellular matrix: unexpected link between mitotic focal adhesion and endocytic plaques.

I suggest to label the reticular adhesion in figure 3 B with the abbreviation RA introduced in the text.

As it is not yet completely established whether all these structures (e.g. the mitotic-resistant adhesion complexes, the reticular adhesions, the coated plaques) are indeed the same type of structures, or specialized subtypes enriched in specific regions of the cell and/or in specific cellular contexts, we have preferred to refer to these structures more generically as ‘mitotic focal adhesion complexes’. We have however added a sentence to explain better their relationship in the legend of Figure 4.